# Associations between Chest CT Abnormalities and Clinical Features in Patients with the Severe Fever with Thrombocytopenia Syndrome

**DOI:** 10.3390/v14020279

**Published:** 2022-01-28

**Authors:** Hiroki Ashizawa, Kazuko Yamamoto, Nobuyuki Ashizawa, Kazuaki Takeda, Naoki Iwanaga, Takahiro Takazono, Noriho Sakamoto, Makoto Sumiyoshi, Shotaro Ide, Asuka Umemura, Masataka Yoshida, Yuichi Fukuda, Tsutomu Kobayashi, Masato Tashiro, Takeshi Tanaka, Shungo Katoh, Konosuke Morimoto, Koya Ariyoshi, Shimpei Morimoto, Mya Myat Ngwe Tun, Shingo Inoue, Kouichi Morita, Shintaro Kurihara, Koichi Izumikawa, Katzunori Yanagihara, Hiroshi Mukae

**Affiliations:** 1Department of Respiratory Medicine, Nagasaki University Graduate School of Biomedical Sciences, Nagasaki 852-8102, Japan; hiroashi@nagasaki-u.ac.jp (H.A.); hmukae@nagasaki-u.ac.jp (H.M.); 2Department of Respiratory Medicine, Nagasaki University Hospital, Nagasaki 852-8102, Japan; n-ashizawa@nagasaki-u.ac.jp (N.A.); k-takeda@nagasaki-u.ac.jp (K.T.); niwanaga@nagasaki-u.ac.jp (N.I.); takahiro-takazono@nagasaki-u.ac.jp (T.T.); nsakamot@nagasaki-u.ac.jp (N.S.); 3Department of Infection Control and Education Center, Nagasaki University Hospital, Nagasaki 852-8102, Japan; mtashiro@nagasaki-u.ac.jp (M.T.); ttakeshi@nagasaki-u.ac.jp (T.T.); koizumik@nagasaki-u.ac.jp (K.I.); 4Department of Respiratory Medicine, Isahaya General Hospital, Isahaya 854-8501, Japan; maksumiyoshi@gmail.com (M.S.); side.ngs@gmail.com (S.I.); 5Department of Respiratory Medicine, Sasebo City General Hospital, Sasebo 857-8511, Japan; asminematsu@gmail.com (A.U.); mstkism1.15@gmail.com (M.Y.); kazunon2007@gmail.com (Y.F.); 6Department of Respiratory Medicine, Sasebo Chuo Hospital, Sasebo 857-1195, Japan; tsutomu.kobayashi@mac.com; 7Department of General Internal Medicine, Nagasaki Rosai Hospital, Nagasaki 857-0134, Japan; shun_5@hotmail.com; 8Department of General Internal Medicine and Clinical Infectious Diseases, Fukushima Medical University, Fukushima 960-1295, Japan; 9Department of Clinical Medicine, Institute of Tropical Medicine, Nagasaki University, Nagasaki 852-8523, Japan; komorimo@nagasaki-u.ac.jp (K.M.); kari@nagasaki-u.ac.jp (K.A.); 10Clinical Research Center, Nagasaki University Hospital, Nagasaki 852-8102, Japan; morimoto.s@nagasaki-u.ac.jp; 11Department of Virology, Institute of Tropical Medicine, Nagasaki University, Nagasaki 852-8523, Japan; myamyat@tm.nagasaki-u.ac.jp (M.M.N.T.); samakirice2012@gmail.com (S.I.); moritak@nagasaki-u.ac.jp (K.M.); 12Department of Medical Safety, Nagasaki University Hospital, Nagasaki 852-8102, Japan; kurihiro@nagasaki-u.ac.jp; 13Department of Laboratory Medicine, Nagasaki University Hospital, Nagasaki 852-8102, Japan; k-yanagi@nagasaki-u.ac.jp

**Keywords:** severe fever with thrombocytopenia syndrome, lung abnormalities, chest computed tomography, ground-glass opacity

## Abstract

Severe fever with thrombocytopenia syndrome (SFTS) is an emerging infectious disease caused by the SFTS virus. It involves multiple organ systems, including the lungs. However, the significance of the lung involvement in SFTS remains unclear. In the present study, we aimed to investigate the relationship between the clinical findings and abnormalities noted in the chest computed tomography (CT) of patients with SFTS. The medical records of 22 confirmed SFTS patients hospitalized in five hospitals in Nagasaki, Japan, between April 2013 and September 2019, were reviewed retrospectively. Interstitial septal thickening and ground-glass opacity (GGO) were the most common findings in 15 (68.1%) and 12 (54.5%) patients, respectively, and lung GGOs were associated with fatalities. The SFTS patients with a GGO pattern were elderly, had a disturbance of the conscious and tachycardia, and had higher c-reactive protein levels at admission (*p* = 0.009, 0.006, 0.002, and 0.038, respectively). These results suggested that the GGO pattern in patients with SFTS displayed disseminated inflammation in multiple organs and that cardiac stress was linked to higher mortality. Chest CT evaluations may be useful for hospitalized patients with SFTS to predict their severity and as early triage for the need of intensive care.

## 1. Introduction

Severe fever with thrombocytopenia syndrome (SFTS) is an emerging tick-borne infectious disease caused by a novel bunyavirus, the SFTS virus (SFTSV), which was first identified in China in 2011 [1]. As it continues to spread, SFTS is attracting increasing global attention, as relatively little is known about this disease, and no standardized treatment currently exists. In 2015, the World Health Organization compiled a list of priority pathogens with the potential to generate an international public health emergency, for which no insufficient, preventive, or curative solutions existed [2]. SFTS was subsequently placed on the list after an annual review in 2016 [3]. In Japan, SFTS was first identified in 2013 [4], and it has been designated as a Category IV disease since 2013 under Japan’s Infectious Diseases Control Law, mandating that physicians notify the authorities of all laboratory-confirmed cases [5]. SFTS exhibits high case fatality rates (CFRs) in humans (13–35%), and from national surveillance data, was found to be especially severe in people over 50 years of age, causing a higher fatality [6], and thus posing a particular threat to Japanese rapidly aging population. So far, SFTS cases have been mainly restricted to localities in the southwestern parts of the Japanese islands [7,8], similar to the distribution of Japanese spotted fever, another tick-borne disease [8].

SFTS involves multiple organs, including the liver, muscles, central nervous system, kidney, genital organs, and lungs [9,10,11]. It was reported that ∼9.6% to 28.7% of patients with SFTS experience respiratory symptoms, including cough, sputum, and dyspnea, with radiographic abnormalities being present in 29 to 45% of patients with SFTS [12]. However, there are limited data regarding the detailed lung involvement in SFTS [12,13,14], with no the studies discussing the relationship between lung imaging abnormalities and the clinical characteristics of patients with the SFTS. In this study, we conducted a retrospective investigation into the relationship between clinical findings and chest CT abnormalities in patients with SFTS that were identified in Nagasaki, a southwestern prefecture of Japan.

## 2. Materials and Methods

### 2.1. Study Population and Clinical Data

We performed a retrospective observational study of adult patients aged ≥ 20 years who were diagnosed with SFTS using a SFTSV PCR test, hospitalized, and underwent chest radiography and CT during the first three days of admission. The patients were registered at five institutions (Nagasaki University Hospital, Isahaya General Hospital, Sasebo City General Hospital, Sasebo Chuo Hospital, and Nagasaki Rosai Hospital) in Nagasaki Prefecture, Japan, between April 2013 and September 2019. Patient medical charts were reviewed retrospectively, and data on baseline characteristics, clinical presentations, laboratory findings, and radiological findings at admission were collected from the medical records. The in-hospital complications and outcomes were also assessed. The disseminated intravascular coagulation (DIC) score was calculated using the Japanese Association for Acute Medicine (JAAM) DIC scoring system [15]. The study protocol was approved by the Institutional Review Board of the Nagasaki University Hospital (approval number, 18121024), Isahaya General Hospital (approval number, 2020-21), Sasebo City General Hospital (approval number, 2020-A024), Sasebo Chuo Hospital (approval number, 2020-17), and Nagasaki Rosai Hospital (approval number: 02003).

### 2.2. SFTSV Real-Time PCR

Sera samples from patients with SFTS, collected at the Nagasaki University Hospital and Sasebo City General Hospital, were measured for the SFTSV viral load using a real-time PCR assay. The total RNA was extracted using a RNeasy Lipid Tissue Mini Kit (Qiagen, Hilden, Germany). The SFTSV-specific primers and probes were designed based on the RdRp region of the consensus sequence of the L segment. The forward primer was SFTS_QPCR_965F: 5′-GCRAGGAGCAACAARCAAACATC-3′, the reverse primer was SFTS_QPCR_1069R: 5′-GCCTGAGTCGGTCTTGATGTC-3′, and the PrimeTimes qPCR probe was FAM/5′-CTCCCRCCC-3′/ZEN/5′-TGGCTACCAAAGC-3′/IBFQ (Integrated DNA Technologies, Coralville, IA, USA) [16]). A real-time RT-PCR was performed using a One Step PrimeScript RT-PCR Kit (Takara Bio Inc., Shiga, Japan) and a 7500 Real-time RT-PCR System (Applied Biosystems). Copy numbers were calculated as the ratio of the copy numbers of the standard controls.

### 2.3. Evaluation of the Chest Radiograph and CT

Two Japanese Respiratory Society (JRS) board-certified pulmonologists, who had 15 and 8 years of experience, respectively, reviewed all chest radiographs and CT examinations independently and came to a consensus. Any disagreements with findings between the two readers were evaluated by a third reader with 15 years of experience as a JRS board-certified pulmonologist. Cardiomegaly was defined as a cardiothoracic (CTR) ratio > 0.50, based on posterior–anterior (PA) chest radiographs or a CTR ratio > 0.55 in anterior–posterior (AP) chest radiographs [17]. The abnormalities in chest CT were characterized by consolidation, ground-glass opacity (GGO), centrilobular nodules, interstitial septal thickening, and bronchial wall thickening. The presence of mediastinal lymph node enlargement (>10 mm along the short axis), pleural effusion, pericardial effusion (pericardial thickness of 4 mm or more [18]), hepatomegaly (diameter of >16.0 cm at craniocaudal line [19]), splenomegaly (width measurement of >10.5 cm [20]), and additional lung findings were also recorded.

### 2.4. Endpoint

The primary endpoint was the association between abnormal chest CT findings and the clinical characteristics of patients with SFTS. First, abnormal CT findings, which correlated with in-hospital fatality in patients with SFTS, were determined. Second, the underlying condition and clinical findings at admission, such as physical symptoms, vital signs, laboratory data, DIC scores, and qSOFA scores were evaluated for associations with abnormal chest CT findings connected to fatality in patients with SFTS. Finally, in-hospital complications and outcomes, such as in-hospital secondary infections (bacterial and/or fungal), ICU admission, and length of hospital stay, were evaluated for their association with abnormal chest CT findings related to fatality.

### 2.5. Statistical Analysis

The results are expressed as means ± standard deviation, medians, and as percentages. SPSS version 25.0 (SPSS, Inc., Chicago, IL, USA) was used to analyze the data. The categorical variables were analyzed using the Chi-square test or Fisher’s exact test, and the continuous variables were analyzed using the Mann–Whitney U test. All the tests were two-tailed and the differences were considered significant at a *p*-value of <0.05.

## 3. Results

### 3.1. Chest CT Findings in Fatal and Non-Fatal Hospitalized Patients with SFTS

A total of 24 hospitalized patients with SFTS during the study period were initially reviewed. Of these patients, two who did not undergo CT examinations were excluded. Finally, the findings were analyzed in 22 hospitalized patients with SFTS in the present study. All the patients were confirmed to be SFTSV-positive using RT-PCR. The chest CT scans were performed in the supine position with 1 mm (*n* = 17), 2 mm (*n* = 3), 3 mm (*n* = 1), or 5 mm (*n* = 1) slices. Chest CT scans were performed in these patients at an average of 4.95 days (range, 1–21 days) following symptom onset, and at an average of 0.14 days (range, 0–1 days) after admission (Table 1). Nineteen patients (86.4%) had abnormal chest CT findings. Interstitial septal thickening and GGO were the most frequently found chest CT abnormalities in 15 (68.1%) and 12 (54.5%) patients, respectively, followed by centrilobular nodules, bronchial wall thickening, and cardiomegaly, which were found in eight (36.4%) patients (Table 1). Hepatomegaly and splenomegaly were found in 27.3 and 13.6%, respectively, of the SFTS patients. Six patients (27.2%) died during hospitalization with the mean duration from disease onset until death being 17.2 days (range: 6–48 days). With regard to the relationship between chest CT abnormalities and fatality in patients with SFTS, GGO was the only finding that was significantly related to death (*p* = 0.015, Table 1).

The typical GGO findings in patients with SFTS are shown in Figure 1. The patchy GGO shadows in the lungs worsened after a week of admission. Chest CT was followed up during hospitalization in seven SFTS patients, including six with GGO and one without GGO at admission. The SFTS patients without GGO at admission did not develop GGO during hospitalization for 30 days. In the SFTS patients with GGO findings at admission, GGO areas peaked at 10.8 (±3.4) days after admission, and these were prolonged until 27.0 (±9.0) days after admission. Table 2 shows the correlation between the GGOs and other abnormal chest CT findings in patients with SFTS. The GGO did not correlate with interstitial septal thickening or other findings, except for cardiomegaly (*p* = 0.031, Table 2). We considered that the GGO findings on the chest CT may correlate with the outcomes in patients with SFTS; therefore, we further investigated the clinical characteristics of patients with SFTS with or without GGO findings on chest CT.

### 3.2. Patient Characteristics and Complications during Hospitalization

Table 3 shows the baseline characteristics and clinical symptoms in the patients with SFTS. The mean age of the patients was 71.4 ± 9.9 years and SFTS was more common in men (68.1%). The GGO was significantly more common in elderly patients (*p* = 0.009). Nineteen (86.3%) patients were farmers and hunters, living or working in wooded and hilly areas before the onset of disease. Seventeen (77.2%) patients had at least one underlying illness, and six (27.2%) patients were smokers. Including chronic lung, cardiovascular, and kidney diseases, the presence of comorbid conditions was comparable between the groups with or without the GGO pattern.

With regard to general symptoms, fever (81.8%) and fatigue (72.7%) were most frequently observed in patients with SFTS, while respiratory symptoms were present in 40.9% of the patients with dyspnea (31.8%) being the most frequent symptom. Respiratory symptoms, including cough and sputum, were comparable between the patients with and those without GGO patterns. Gastrointestinal symptoms were present in 81.8% of the patients with SFTS, with diarrhea being the most common symptom, which was significantly present in the patients without a GGO pattern. Skin rash and tick bites were found in >30% of the patients, and lymphadenopathy was present in 45.5% of the patients with SFTS.

Secondary bacterial or fungal infections occurred in five patients (22.7%) with SFTS during hospitalization. Fungal infections (invasive pulmonary aspergillus, candidemia, and disseminated trichosporonosis) occurred in three patients (13.6%) who presented with GGO in the lung, although the difference was not statistically significant. The length of hospital stay and intensive care unit (ICU) admission also tended to be higher in patients with GGO, although the difference was not statistically significant.

### 3.3. Laboratory Findings

Table 4 shows the vital signs and laboratory findings at admission. Impaired consciousness was observed in 63.6% of patients with SFTS and was significantly present in those with GGO in the lung (*p* = 0.006). The mean body temperature at admission was 38.3 °C in patients with SFTS and a significantly high heart rate (mean, 93.3/min) was observed in those with GGO findings (*p* = 0.002). There were 11 (50.0%) patients with hypoxemia at admission, although there were no differences in the SpO_2_ or SpO_2_/FiO_2_ ratios between the two groups. The qSOFA scores in patients with SFTS between the groups with or without GGO in the lung were comparable.

Leukocytopenia and thrombocytopenia were also observed and were similar between the groups. Abnormal liver function tests, elevated creatine kinase and ferritin levels, renal dysfunction, and coagulation disturbances were generally observed in patients with SFTS, and these were comparable between groups. However, elevated c-reactive protein (CRP) levels were significantly observed in patients with GGO patterns (*p* = 0.038). The mean SFTSV viral titer was higher in patients with GGO patterns, although the difference was not statistically significant. SFTSV serial viral load was measured during hospitalization in nine patients (seven SFTS patients with GGO and two without GGO, Figure 2). Viremia was prolonged for between a week through 4 weeks after admission, though a trend of SFTS patients with GGO was not definable.

## 4. Discussion

Since the first report of SFTS in Japan, in 2013, approximately 640 cases have been reported as of July 2021, with a high mortality rate (12%): 80 deaths [21]. In the present study, we found that GGOs were identified frequently in the lungs of patients with SFTS and correlated with poor outcomes. In a previous report from China, in 2013, chest radiograph (CR) abnormalities were found in 45% of 98 cases diagnosed with SFTS [13], which was lower than the figures in our report, and may be due to the differences in the sensitivities between CR and CT. These were supported by our data, which showed that CR could detect 8 out of 12 (66.6%) GGO findings compared to CT. Another report from Korea found abnormal chest CT findings in 62% of 21 hospitalized patients with SFTS, with a variable consolidation; GGOs were the most commonly identified abnormality (48%), which was consistent with the findings of the present study. From the viewpoint of pathogenesis, GGO in the lungs of patients with SFTS may be generated by lung edema [22] due to accelerated inflammation or disseminated SFTSV in the lung. While the initial CR may have been normal, bilateral pulmonary infiltration indicated the development of pulmonary edema [22]. Although serial follow-up CT was performed in only 7 (31.8%) patients in our study, GGO in the lungs of patients with SFTS became more prominent at approximately 1 to 2 weeks of onset (Figure 1) and was prolonged for a month. In our study, the GGO was correlated with tachycardia, indicating that cardiac stress may result in the generation of GGO in the lungs. The cardiomegaly that was found in 36.4% of our cases was more frequently reported in hospitalized patients with SFTS (90%) [12], which may support this hypothesis. The SFTSV viral titer was not correlated significantly with GGO abnormalities in our study; however, a quantitative RT-PCR was performed partially (72.7%) in two hospitals, and serial SFTSV follow-up measurement was performed in only 40.9% of all patients with SFTS; therefore, it was difficult to make a conclusion regarding this connection. Viral dissemination in the pulmonary alveoli was confirmed using bronchoalveolar lavage in one of our patients who presented with GGO in the lung [11], suggesting that the GGO pattern may be generated by viral dissemination and accelerated inflammation in the lung. Pleural effusion, pericardial effusion, and mediastinal lymph node enlargement in the present study (22.7%, 18.1%, and 13.6%, respectively) were similar to those in a previous study (38%, 24%, and 14%, respectively) [12]. Although SFTS causes various systemic symptoms, there is a significant disturbance in conscious among patients with GGO, which may indicate the risk of the dissemination of SFTS disease to multiple organs [10]. In addition, CRP levels were elevated significantly in patients with GGO, which may indicate an association with systemic inflammation.

The present study had several limitations that are characteristic of retrospective studies. First, symptoms and laboratory data may not have been comprehensively recorded. Second, patients with milder illnesses may not have presented at the hospital. Third, selection bias toward more severe disease with SFTS may have been included in the present study because the chest CT scans were included in the inclusion criteria. Fourth, the sample size may not have been large enough to draw a conclusion on an association between abnormal findings in chest CT and the clinical features in patients with SFTS.

Despite these limitations, to the best of our knowledge, the present study was the first to evaluate the significance of lung abnormalities in chest CT findings in patients with SFTS. Patients with SFTS showing GGO patterns in the lungs, may develop more severe diseases and require intensive care; therefore, attention to abnormal chest findings is important at the time of admission.

## 5. Conclusions

Abnormal lung GGOs were associated with fatalities in patients with SFTS. Pulmonary GGO findings were accompanied frequently with cardiomegaly and were mainly found in elderly patients with SFTS. Patients with SFTS with GGO findings in the lung often present with impaired consciousness, tachycardia, and high inflammation at admission. These results suggested that the GGO patterns in patients with SFTS had a more disseminated inflammation in multiple organs, and that cardiac stress was linked to a higher mortality. Chest CT evaluations may be useful for hospitalized patients with SFTS, in order to predict the severity of their condition and the need for early triage to meet their need for intensive care. Further studies are needed to determine the clinical impact of chest CT evaluations on the diagnosis and treatment of SFTS.

## Figures and Tables

**Figure 1 viruses-14-00279-f001:**
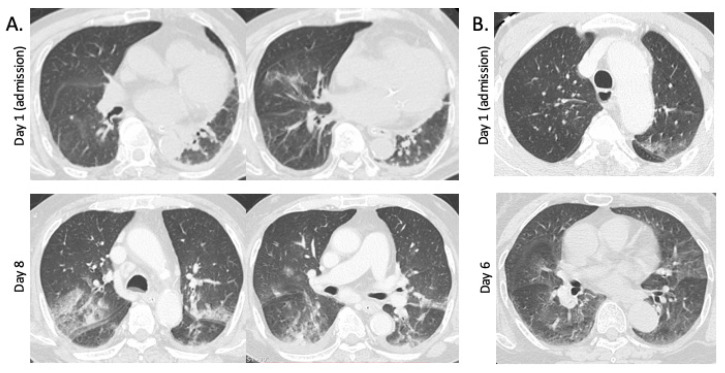
Ground-glass opacities (GGOs) in chest CT in patients with SFTS. (**A**) Chest CT images in a 79-year-old nonfatal male patient. Focal GGOs in the lower lobes of the lungs, interstitial septal thickening, cardiomegaly, and pleural effusion were found at admission (2 days after onset). At day 8 of admission, multifocal GGOs with patchy consolidations are present in both lungs. (**B**) Chest CT images of a 73-year-old fatal female patient. Focal GGO is present at upper lobe of left lung at admission (3 days after onset). Multifocal GGOs, interstitial septal thickening, and cardiomegaly are found at day 6 of admission.

**Figure 2 viruses-14-00279-f002:**
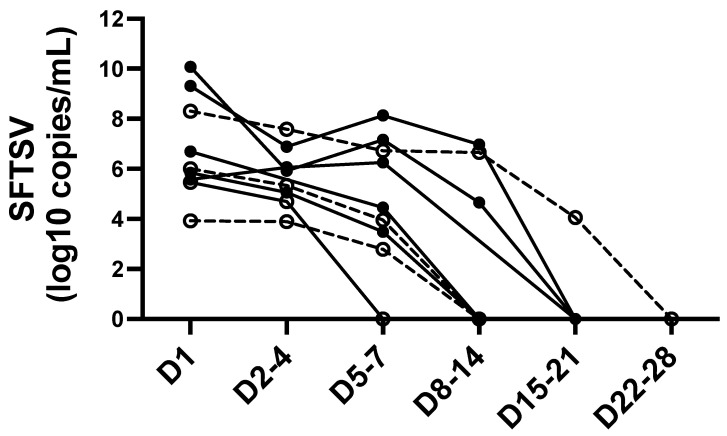
Serial SFTSV viral load in patients with SFTS. Open circles with dotted lines represent the SFTS patients without GGO in chest CT at admission. Filled circles with solid lines represent the SFTS patients with GGO in chest CT at admission. Days (D) after admission are displayed on X-axis. Abbreviations: SFTSV, severe fever with thrombocytopenia syndrome virus.

**Table 1 viruses-14-00279-t001:** Chest CT findings in fatal and non-fatal hospitalized patients with SFTS.

	All (*n* = 22)	Fatal (*n* = 6)	Non-Fatal (*n* = 16)	*p*-Value
Time from symptoms onset to chest CT, days ± SD	4.95 ± 4.09	5.17 ± 0.98	4.94 ± 1.16	0.910
Time from admission to chest CT, days ± SD	0.14 ± 0.35	0.00 ± 0.00	0.25 ± 0.11	0.041
Abnormal chest CT findings	19 (86.4)	6 (100)	13 (81.3)	0.532
Consolidation	5 (22.7)	2 (33.3)	3 (18.8)	0.585
Ground-glass opacity	12 (54.5)	6 (100)	6 (37.5)	0.015
Interstitial septal thickening	15 (68.1)	6 (100)	9 (56.3)	0.121
Centrilobular nodule	8 (36.4)	2 (33.3)	6 (37.5)	1.000
Bronchial wall thickening	8 (36.4)	2 (33.3)	6 (37.5)	1.000
Cardiomegaly	8 (36.4)	3 (50.0)	5 (31.3)	0.624
Pleural effusion	5 (22.7)	3 (50.0)	2 (12.5)	0.100
Pericardial effusion	4 (18.2)	1 (16.7)	3 (18.8)	1.000
Mediastinal lymph node enlargement	3 (13.6)	1 (16.7)	2 (12.5)	1.000
Hepatomegaly	6 (27.3)	1 (16.7)	5 (31.3)	0.634
Splenomegaly	3 (13.6)	0 (0.0)	3 (18.8)	0.532

Data are no. (%) of patients. Abbreviations: CT, computed tomography; SD, standard deviation; SFTS, severe fever with thrombocytopenia syndrome.

**Table 2 viruses-14-00279-t002:** Correlation of GGO and other abnormal chest CT findings in patients with SFTS.

	All (*n* = 22)	GGO (+) (*n* = 12)	GGO (−) (*n* = 10)	*p*-Value
Consolidation	5 (22.7)	3 (25.0)	2 (20.0)	1.000
Interstitial septal thickening	15 (68.2)	9 (75.0)	6 (60.0)	0.652
Centrilobular nodule	8 (36.4)	3 (25.0)	5 (50.0)	0.378
Bronchial wall thickening	8 (36.4)	4 (33.3)	4 (40.0)	1.000
Cardiomegaly	8 (36.4)	7 (58.3)	1 (10.0)	0.031
Pleural effusion	5 (22.7)	4 (33.3)	1 (10.0)	0.323
Pericardial effusion	4 (18.2)	3 (25.0)	1 (10.0)	0.594
Mediastinal lymph node enlargement	3 (13.6)	3 (25.0)	0 (0.0)	0.221
Hepatomegaly	6 (27.3)	4 (33.3)	2 (20.0)	0.646
Splenomegaly	3 (13.6)	3 (25.0)	0 (0.0)	0.221

Data are no. (%) of patients. Abbreviations: GGO, ground-glass opacity; CT, computed tomography; SFTS, severe fever with thrombocytopenia syndrome.

**Table 3 viruses-14-00279-t003:** Underlying conditions, symptoms, and outcomes in patients with SFTS.

	All (*n* = 22)	GGO (+) (*n* = 12)	GGO (−) (*n* = 10)	*p*-Value	95% CI
Age, ±SD	71.4 ± 9.9	76.2 ± 1.9	65.6 ± 3.2	0.009	2.98–18.15
Male gender	15 (68.1)	8 (66.7)	7 (70.0)	1.000	
Smoking	6 (27.2)	2 (16.7)	4 (40.0)	0.348	
Farmers, hunters, living or working in wooded and hilly areas	19 (86.3)	9 (41.6)	10 (100)	0.221	
Underlying diseases	17 (77.3)	11 (75.0)	6 (60.0)	0.135	
Chronic lung diseases	8 (36.4)	5 (41.6)	3 (30.0)	0.675	
Cardiovascular diseases	12 (54.5)	7 (58.3)	5 (50.0)	1.000	
Chronic kidney diseases	2 (9.0)	1 (8.3)	1 (10.0)	1.000	
Diabetes mellitus	4 (18.1)	2 (16.6)	2 (20.0)	1.000	
Cancer	2 (9.0)	1 (8.3)	1 (10.0)	1.000	
General symptoms					
Fatigue	16 (72.7)	9 (75.0)	7 (70.0)	1.000	
Fever	18 (81.8)	9 (75.0)	9 (90.0)	0.594	
Chill	2 (9.1)	0 (0.0)	2 (20.0)	0.195	
Headache	4 (18.1)	0 (0.0)	4 (40.0)	0.029	
Dizziness	3 (13.6)	2 (16.7)	1 (10.0)	1.000	
Myalgia	1 (4.5)	1 (8.3)	0 (0.0)	1.000	
Respiratory symptoms	9 (40.9)	6 (50.0)	3 (30.0)	0.415	
Cough	1 (4.5)	0 (0.0)	1 (10.0)	0.455	
Sputum	2 (9.1)	2 (16.6)	0 (0.0)	0.481	
Dyspnea	7 (31.8)	5 (41.6)	2 (20.0)	0.381	
Gastrointestinal symptoms	18 (81.8)	8 (66.7)	10 (100)	0.096	
Nausea	3 (13.6)	1 (8.3)	2 (20.0)	0.571	
Anorexia	4 (18.1)	3 (25.0)	1 (10.0)	0.594	
Diarrhea	11 (50.0)	3 (25.0)	8 (80.0)	0.030	
Skin and other symptoms					
Skin rash	8 (36.4)	5 (41.6)	3 (30.0)	0.675	
Tick bite	7 (31.8)	4 (33.3)	3 (30.0)	1.000	
Lymphadenopathy	10 (45.5)	6 (50.0)	4 (40.0)	0.691	
Complications and outcomes					
Secondary infection	5 (22.7)	4 (33.3)	1 (10.0)	0.323	
Fungal infection	3 (13.6)	3 (25.0)	0 (0.0)	0.221	
Length of hospital stay (days)	35.2	46.7	21.4	0.329	−27.41–77.95
ICU admission	7 (31.8)	6 (50.0)	1 (10.0)	0.074	
Length of ICU stay (days)	7.1	12.3	0.8	0.061	−0.65–23.55

Data are mean or no. (%) of patients. Abbreviations: SFTS, severe fever with thrombocytopenia syndrome; qSOFA, quick sepsis related organ failure assessment; DIC, disseminated intravascular coagulation; ICU, intensive care unit; SD, standard deviation.

**Table 4 viruses-14-00279-t004:** Vital signs and laboratory data on the admission of patients with SFTS.

	All (*n* = 22)	GGO (+) (*n* = 12)	GGO (−) (*n* = 10)	*p*-Value	95% CI
Vital signs					
Consciousness disturbance	14 (63.6)	11 (91.7)	3 (30.0)	0.006	
Body temperature (°C)	38.3	38.4	38.2	0.721	−0.69–0.98
Systolic blood pressure(mmHg)	127.4	133.1	120.5	0.114	−3.31–28.48
Heart rate (/min)	84.8	93.3	74.6	0.002	7.54–29.92
SpO_2_ (%)	96.5	96.3	96.7	0.695	−2.29–1.56
SpO_2_/FiO_2_ ratio	412.8	401.5	426.4	0.494	−99.66–49.74
qSOFA score	0.96	1.17	0.70	0.191	−0.22–1.15
Rales	4 (18.1)	3 (25.0)	1 (10.0)	0.594	
Laboratory data					
WBC (/μL)	1725.5	1733.3	1716.0	0.967	−844.85–879.52
Neutrophil (%)	64.6	65.3	63.7	0.816	−12.33–15.48
Lymphocyte (%)	26.7	26.6	26.8	0.961	−9.38–8.95
Atypical lymphocyte (%)	2.09	1.92	2.30	0.803	−3.55–2.78
Platelet (×10^3^/μL)	52.6	50.9	54.6	0.781	−30.83–23.5
Hb (g/dL)	14.0	13.9	14.3	0.636	−2.24–1.4
AST (IU/L)	342.4	404.6	267.8	0.311	−137.6–411.17
ALT (IU/L)	117.8	128.8	104.5	0.506	−50.62–99.29
LDH (IU/L)	830.4	927.5	713.8	0.242	−156.39–583.79
CK (IU/L)	2560.6	2670.4	2439.8	0.908	−3900.54–4361.67
CRP (mg/dL)	0.86	1.34	0.27	0.038	0.070–2.07
PCT (ng/mL)	0.27	0.37	0.19	0.143	−0.07–0.44
Alb (g/dL)	3.24	3.21	3.28	0.704	−0.46–0.32
Ferritin (ng/mL)	20,194.8	25,839.5	12,433.3	0.536	−31,373.2–58,185.7
BUN (mg/dL)	26.8	25.2	28.7	0.639	−19.08–11.99
Cre (mg/dL)	1.49	1.70	1.25	0.649	−1.57–2.46
eGFR (mL/min/1.73 m^2^)	57.3	61.3	52.5	0.404	−12.68–30.2
Na (mEq/L)	132.3	130.6	134.4	0.138	−9.01–1.38
K (mEq/L)	4.00	4.02	3.96	0.824	−0.47–0.58
Cl (mEq/L)	99.6	97.4	102.3	0.084	−10.48–0.71
APTT (%)	48.7	52.2	44.6	0.240	−5.51–20.74
PT-INR	1.06	1.07	1.04	0.442	−0.056–0.12
D-dimer (μg/mL)	14.4	18.4	9.8	0.211	−5.35–22.55
FDP (μg/mL)	27.6	32.7	20.9	0.365	−14.79–38.37
DIC score	4.27	4.92	3.50	0.108	−0.34–3.17
SFTSV viral load (copies/mL)	8.76 × 10^8^	1.65 × 10^9^	1.03 × 10^8^	0.390	−1.63 × 10^9^–4.73 × 10^9^
Proteinuria	21(95.5)	12 (100)	9 (90.0)	0.455	
Hematuria	19(86.4)	10 (83.3)	9 (90.0)	1.000	

Data are presented as mean. Abbreviations: WBC, white blood cells; Hb, hemoglobin; AST, aspartate aminotransferase; ALT, alanine transaminase; LDH, lactate dehydrogenase; CK, creatine kinase; CRP, c-reactive protein; PCT, procalcitonin; Alb, albumin; BUN, blood urea nitrogen; Cre, creatinine; eGFR, estimated glomerular filtration rate; APTT, activated partial thromboplastin time; PT-INR, prothrombin time test and INR; FDP, fibrinogen-degradation products; DIC, disseminated intravascular coagulation; SFTSV, severe fever with thrombocytopenia syndrome virus.

## Data Availability

Data are contained within the article.

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
