# Peer review of "Associations between Chest CT Abnormalities and Clinical Features in Patients with the Severe Fever with Thrombocytopenia Syndrome"

_viruses, 2022, doi:10.3390/v14020279_

Round 1
Reviewer 1 Report
Authors showed that “Chest CT evaluations may be useful for hospitalized patients with SFTS to predict their severity and as early triage for their need for intensive care”.
Could authors consider the following?
For sample size:
Total sample is 22 (fatal n=6 and nonfatal n=16) in this study and 22 is small and I worry about sample size for extraction of concrete results.
For clinical course of SFTS patient study:
For using useful for hospitalized patients with SFTS to predict their severity and as early triage for their need for intensive care, I recommend that authors show chest CT evaluations data according to clinical course of SFTS patient and also compare with SFTSV real-time PCR according to clinical course.
Author Response
Reviewer: 1
Comments and Suggestions for Authors
Authors showed that “Chest CT evaluations may be useful for hospitalized patients with SFTS to predict their severity and as early triage for their need for intensive care”.
Thank you for reviewing our paper and making insightful comments. We have revised our manuscript to the best of our ability according to your suggestions. Please find our point-by-point responses below.
[C1] Could authors consider the following?
For sample size:
Total sample is 22 (fatal n=6 and nonfatal n=16) in this study and 22 is small and I worry about sample size for extraction of concrete results.
(Response 1) Thank you for your comment and suggestion. We agree that a sample size of 22 is small to extract concrete results, which address in the limitations at Discussion section (lines 308-309). Since SFTS is not a common disease and the number of SFTS patients with available chest CT data are even lower as those in a previous report [11], it is difficult to include more patients in our study. We believe that our paper gives a new clinical insight to the field of this disease even though the overall number of included cases are low.
[C2] For clinical course of SFTS patient study:
For using useful for hospitalized patients with SFTS to predict their severity and as early triage for their need for intensive care, I recommend that authors show chest CT evaluations data according to clinical course of SFTS patient and also compare with SFTSV real-time PCR according to clinical course.
(Response 2) Thank you for your suggestion. A Figure 2 was added to display serial SFTSV viral load in the clinical course of the SFTS patients, with or without GGO in chest CT findings. Description of those was added to the Results section (lines 304-307) and Discussion section (lines 355-356). Moreover, the course of GGO findings in the serial chest CT is described in lines 195-200 in the Results rection, and in lines 343-345 in the Discussion section, in which the development and persistence of GGO findings in the SFTS patients are clarified.
Reviewer 2 Report
The manuscript by Ashizawa and colleagues describes the clinical findings and chest CT findings in Japanese patients infected with severe fever with thrombocytopenia syndrome virus (SFTSV) between 2013 and 2019. The report is well and clearly written. Even though the overall number cases is rather low, the report provides interesting new insight into the disease.
A minor comment regarding the Introduction, I think it would be fair to mention that the disease is tick-borne. Are there more ticks in the southwestern parts of the Japanese islands or is the behaviour of habitants different in those parts? Or is there something else that might explain the difference in case numbers? Why do the authors speculate SFTSV to "posing a particular threat to Japan’s rapidly aging population", if the virus indeed is tick-borne?. Just curious.
Author Response
Reviewer: 2
Comments and Suggestions for Authors
The manuscript by Ashizawa and colleagues describes the clinical findings and chest CT findings in Japanese patients infected with severe fever with thrombocytopenia syndrome virus (SFTSV) between 2013 and 2019. The report is well and clearly written. Even though the overall number cases is rather low, the report provides interesting new insight into the disease.
Thank you for reviewing our paper. We appreciate your positive feedback. Please find our responses below, showing our compliance with your valuable suggestions.
[C1] A minor comment regarding the Introduction, I think it would be fair to mention that the disease is tick-borne. Are there more ticks in the southwestern parts of the Japanese islands or is the behaviour of habitants different in those parts? Or is there something else that might explain the difference in case numbers? Why do the authors speculate SFTSV to "posing a particular threat to Japan’s rapidly aging population", if the virus indeed is tick-borne?. Just curious.
(Response 1) Thank you for helping us improve our manuscript. Please find “a tick-borne infectious disease” now included as the first line in the Introduction (lines 62-63). A reference [22] was added to refer the distribution of SFTS patients compared to those with other tick-borne infectious diseases in Japan. Lines 76-77 were added to explain these. In addition, Line 74 was modified to clearly state that elderly had higher fatality rate among the SFTS patients in Japan.
Round 2
Reviewer 1 Report
We believe that our paper gives a new clinical insight to the field of this disease even though the overall number of included cases are low.
: I hope that the authors are continuing to collect samples for good results in his or her further study.